# A Self-Supplying H_2_O_2_ Modified Nanozyme-Loaded Hydrogel for Root Canal Biofilm Eradication

**DOI:** 10.3390/ijms231710107

**Published:** 2022-09-03

**Authors:** Jiazhuo Song, Lihua Hong, Xinying Zou, Hamed Alshawwa, Yuanhang Zhao, Hong Zhao, Xin Liu, Chao Si, Zhimin Zhang

**Affiliations:** 1Department of Endodontics, School of Dentistry, Jilin University, Changchun 130021, China; 2Jilin Provincial Key Laboratory of Tooth Development and Bone Remodeling, Changchun 130021, China

**Keywords:** hydrogel, reactive oxygen species, biofilm, root canal therapy

## Abstract

The success of root canal therapy depends mainly on the complete elimination of the root canal bacterial biofilm. The validity and biocompatibility of root canal disinfectant materials are imperative for the success of root canal treatment. However, the insufficiency of the currently available root canal disinfectant materials highlights that more advanced materials are still needed. In this study, a nanozyme-loaded hydrogel (Fe_3_O_4_-CaO_2_-Hydrogel) was modified and analyzed as a root canal disinfectant material. Fe_3_O_4_-CaO_2_-Hydrogel was fabricated and examined for its release profile, biocompatibility, and antibacterial activity against *E. faecalis* and *S. sanguis* biofilms in vitro. Furthermore, its efficiency in eliminating the root canal bacterial biofilm removal in SD rat teeth was also evaluated. The results in vitro showed that Fe_3_O_4_-CaO_2_-Hydrogel could release reactive oxygen species (ROS). Moreover, it showed good biocompatibility, disrupting bacterial cell membranes, and inhibiting exopolysaccharide production (*p* < 0.0001). In addition, in vivo results showed that Fe_3_O_4_-CaO_2_-Hydrogel strongly scavenged on root canal biofilm infection and prevented further inflammation expansion (*p* < 0.05). Altogether, suggesting that Fe_3_O_4_-CaO_2_-Hydrogel can be used as a new effective biocompatible root canal disinfectant material. Our research provides a broad prospect for clinical root canal disinfection, even extended to other refractory infections in deep sites.

## 1. Introduction

Incomplete elimination of the root canal biofilm is one of the main reasons for root canal treatment failure; thus, ensuring complete root canal disinfection before root canal filling is crucial [1,2]. Under the premise of ensuring correct root canal therapy, the mechanical shaping of the root canals plays a significant role in removing infected dentin and giving access to the root canal disinfectant to reach the most root canal system irregularities [3]. Traditional root canal disinfectants such as sodium hypochlorite (NaClO) solution, chlorohexidine (CHX) solution, calcium hydroxide (Ca(OH)_2_) paste, and other drugs have certain drawbacks [4,5]. A high-concentration NaClO has a broad-spectrum antibacterial activity; however, it might cause severe damage to the surrounding tissues [6]. Therefore, a low-concentration NaClO is often used in clinical practice, although it has limited antibacterial activity [7,8]. Studies suggest that CHX may cause tooth discoloration and allergic reactions [9]. Ca(OH)_2_ paste has been regarded as the interappointment medication of choice for its antimicrobial effects and tissue-dissolving capacity. However, Ca(OH)_2_ remnants can jeopardize the bonding ability of root canal sealers and increase apical leakage [10]. Therefore, the development of alternative root canal disinfection materials is highly warranted. 

Nanozymes are a class of nanoparticles (NPs) containing multivalent metal elements with high catalytic activity [11,12]. Among different nanozymes, the most commonly used nanozymes with antibacterial activity are those exhibiting peroxidase activity. Fe_3_O_4_ NPs are the most common nanozymes that decompose H_2_O_2_ through a Fenton-like reaction to generate reactive oxygen species (ROS) [13], indicating that they are an effective alternative antibacterial material [14]. However, a limited H_2_O_2_ concentration in the bacterial microenvironment under hypoxic conditions limits the efficiency of the Fe_3_O_4_ NPs Fenton reaction, resulting in a low ROS output and an unsatisfactory antibacterial effect [15]; furthermore, the addition of exogenous H_2_O_2_ might interfere with tissue healing [16]. Therefore, it is necessary to develop H_2_O_2_-free, efficient and biocompatible agents to treat bacterial infection. Several strategies have been used to address the issue of low H_2_O_2_ concentration in the bacterial microenvironment [17]. CaO_2_ NPs are often used as efficient H_2_O_2_ donors; however, the produced H_2_O_2_ is easily decomposed and detoxified by the enzyme catalase in the bacteria [18]. Therefore, Fe_3_O_4_ and CaO_2_ may be combined to form self-supplying H_2_O_2_ modified nanozymes, resulting in a synergistic reaction to produce a strong antibiofilm effect. In an acidic environment, CaO_2_ can rapidly decompose, and the accumulation of released products might compromise its biocompatibility. Therefore, a suitable carrier should be developed to avoid the explosive decomposition of CaO_2_ [19]. 

Hydrogel materials have good fluidity and their shape can change according to the environment, indicating that they can be an ideal drug carrier [20]. Moreover, hydrogels are characterized by simple preparation, easy availability, low toxicity, convenient use, and controlled release; thus, hydrogel materials have been widely used in the fabrication of various controlled drug release materials such as microspheres, fiber scaffolds, and polymer membranes [21,22]. Natural polymers are more biocompatible than synthetic polymers. Therefore, chitosan (CS) and β-glycerophosphate (GP) are commonly used as synthetic materials for hydrogels in biomedicine. Upon addition of gelatin (GA), CS and GP are cross-linked by electrostatic action, and GA, CS, and GP are adequately biocompatible [23,24]. Use of a hydrogel to encapsulate nanozyme provides a protective shell, which prevents the explosive release of ROS and reduces the toxicity of nanozymes. 

In this study, we synthesized a modified nanozyme-loaded hydrogel (Fe_3_O_4_-CaO_2_-Hydrogel), which generated ROS for root canal biofilm eradication after coming in contact with the bacterial microenvironment without additional excitation conditions. The biocompatibility and antimicrobial properties of this hydrogel were evaluated to confirm whether this hydrogel could be considered as a novel root canal disinfection solution in the future. 

## 2. Results

### 2.1. Characterization of the Modified Nanozyme

As shown in transmission electron microscopy (TEM) images (Figure 1C–E), the synthesized Fe_3_O_4_-CaO_2_ NPs were circular, with diameters in the range of 30–40 nm. XRD images (Figure 1F) show typical peaks of Fe_3_O_4_-CaO_2_ NPs, XPS images (Figure 1G) show obvious Fe, Ca, and O absorption peaks.

### 2.2. Characterization of the Modified Nanozyme-Loaded Hydrogel

At ambient temperature of 37 °C, the hydrogels solidified; the curing time was approximately 60 ± 5 s. To verify the fluidity of the pre-colloid, in a 9-mm dish, “GEL” was written by 5 mL syringes using 21G needle (Figure 1H). SEM showed obvious honeycomb-structured and flaky pore images of the CS-based hydrogel inside the colloid (Figure 1I). The homogenous elemental distribution of Fe, Ca, O, and Pd of the Fe_3_O_4_-CaO_2_-Hydrgel is demonstrated by the SEM mapping (Figure 1J).

### 2.3. Release Profile of ROS from the Hydrogel

The absorbance change in the diphenylisobenzofuran (DPBF) reaction solution at 410 nm is shown in Figure 1L,M. After 168 h, at pH of 7.4 and 5.5, the release ratios of ROS loaded with different concentrations of modified nanozyme hydrogels were 0.40, 0.43, 0.50, 0.57, and 0.66, and 0.37, 0.41, 0.47, 0.54, and 0.60. These results indicated that more ROS were released under acidic conditions, and the amount of ROS generated was proportional to the concentration of the modified nanozymes. In the first 24 h, ROS were rapidly released, which may be important for the fast bacterial killing.

### 2.4. Concentration Selection of the Modified Nanozymes

According to the results of CCK-8 assay (Figure 2A–C), cell viability of all tested groups decreased with an increase in the concentration of modified nanozymes compared with that of the control group; the 2.0 and 2.5 mg/mL groups showed cell viability of only 24.22% (MD: 0.38 ± 0.05, *p* < 0.0001) and 21.26% (MD: 0.35 ± 0.05, *p* < 0.0001), respectively. In the 0.5 and 1.0 mg/mL groups, good cell growth environment was observed on day 3 of incubation, and the cell viability for the 1.5 mg/mL group was 77.48%. According to the International Organization for Standardization (ISO) classification [25], three groups in the safe range were selected, namely 0.5, 1.0, and 1.5 mg/mL. Figure 2D–F shows the growth results of bacterial colonies of *Enterococcus faecalis* and *Streptococcus sanguis* treated with different concentrations of the nanozyme-loaded hydrogels. The three groups with good biocompatibility showed some antibacterial effect against *E. faecalis* and *S. sanguinis*. Bacteriostatic rates against *E. faecalis* for the three groups were 29.91% (MD: 77.33 ± 10.26, *p* < 0.05), 53.78% (MD: 51.00 ± 4.58, *p* < 0.001), and 91.54% (MD: 9.33 ± 4.04, *p* < 0.0001), whereas the corresponding rates against *S. sanguinis* were 44.44% (MD: 10.00 ± 10.58, *p* < 0.01), 83.89% (MD: 29.00 ± 8.19, *p* < 0.001), and 97.96% (MD: 3.67 ± 1.53, *p* < 0.001), The 1.5 mg/mL group showed appropriate biosafety and antibacterial effect; thus, 1.5 mg/mL can be considered as the optimal concentration for loading the modified nano-enzyme in hydrogels.

### 2.5. Antibiofilm In Vitro

The antibiofilm rate of each group was calculated using the plate colony counting method. As shown in Figure 3E–G, the Fe_3_O_4_-CaO_2_-Hydrogel group showed a strong inhibitory effect on *E. faecalis* and *S. sanguis*, and only a few colonies remained on the surface of the solid plate. Compared with that of the control group, the antibiofilm rates of Fe_3_O_4_-CaO_2_-Hydrogel, CaO_2_-Hydrogel, Fe_3_O_4_-Hydrogel, and Hydrogel against *E. faecalis* were 91.97% (MD: 16.67 ± 8.14, *p* < 0.0001), 73.03% (MD: 56.00 ± 6.24, *p* < 0.0001), 35.15% (MD: 134.67 ± 5.03, *p* < 0.001), and 13.48% (MD: 179.67 ± 17.47, *p* > 0.05), respectively, and those against *S. sanguis* were 98.58% (MD: 4.00 ± 1.00, *p* < 0.0001), 92.36% (MD: 21.67 ± 4.04, *p* < 0.0001), 41.48% (MD: 166.00 ± 14.18, *p* < 0.001), and 23.03% (MD: 218.33 ± 36.17, *p* < 0.05), respectively.

The effect of the modified nanozyme-loaded hydrogels on the metabolic activity of multi-strain biofilms was examined. As described in Section 4.5.2, the live/dead staining test was used to evaluate the activity of biofilms grown on the surface of the cover glass. As shown in Figure 3H, most of the root canal biofilm were dead when they were in contact with the Fe_3_O_4_-CaO_2_-Hydrogel sample. Furthermore, the MTT results (Figure 3I) showed that compared with the control group, the biofilm survival rates of Fe_3_O_4_-CaO_2_-Hydrogel, CaO_2_-Hydrogel, Fe_3_O_4_-Hydrogel, and Hydrogel were 16.05% (MD: 0.317 ± 0.03, *p* < 0.0001), 38.89% (MD: 0.77 ± 0.05, *p* < 0.0001), 84.57% (MD: 1.67 ± 0.14, *p* < 0.01), and 97.15% (MD: 1.92 ± 0.15, *p* > 0.05), respectively. Only the Fe_3_O_4_-CaO_2_-Hydrogel group showed strong antibiofilm activity. Crystal violet staining of the biofilm showed the same metabolic activity trend (Figure 3J); the activity from low to high was: Fe_3_O_4_-CaO_2_-Hydrogel < CaO_2_-Hydrogel < Fe_3_O_4_-Hydrogel < Hydrogel < Control. 

The CaO_2_-Hydrogel and Fe_3_O_4_-Hydrogel were not ideal for biofilm scavenging, while the Fe_3_O_4_-CaO_2_-Hydrogel showed fast and strong antibacterial ability, which can be attributed to the occurrence of a Fenton-like reaction. The specific values are provided in Table 1.

### 2.6. Effect of the Modified Nanozyme-Loaded Hydrogel on Bacterial Structure

SEM results (Figure 4A) showed that each group exhibited different extents of damage to the cell membrane structure. Bacteria treated with Fe_3_O_4_-CaO_2_-Hydrogel and CaO_2_-Hydrogel exhibited considerable cell membrane damage and leakage of bacterial contents (as indicated by arrows). For the Fe_3_O_4_-Hydrogel, the bacteria showed depressions on the surface (as indicated by arrows), which may be related to the up-take of Fe_3_O_4_ NPs by the bacteria; in the control and hydrogel groups, a more complete cell membrane morphology was observed. The presence of ROS caused obvious changes in outer membrane permeability, cell membrane function, and leakage of intracellular substances. This conclusion was confirmed through K ion efflux experiments (Figure 4B,C). ICP results showed that the amount of K ion leakage in the Fe_3_O_4_-CaO_2_-Hydrogel and CaO_2_-Hydrogel groups was considerably higher than that in the control groups, proving that the bacterial cell membrane was damaged.

### 2.7. Detection of Extracellular Polysaccharides

The results of quantitative analysis of extracellular polysaccharides (EPS) production are shown in Figure 4D. Compared with the control group, Fe3O4-Hydrogel (96.50%) (MD: 1.08 ± 0.14, *p* > 0.05) and Hydrogel (98.55%) (MD: 1.10 ± 0.18, *p* > 0.05) did not negatively affect polysaccharide production in biofilms; however, polysaccharides were considerably lower in samples treated with CaO_2_-Hydrogel (48.78%) (MD: 0.56 ± 0.14, *p* < 0.001) and Fe_3_O_4_-CaO_2_-Hydrogel (19.85%) (MD: 0.23 ± 0.03, *p* < 0.0001), respectively.

### 2.8. In Vivo Antibiofilm Activity

#### 2.8.1. Imaging Examination

Radiographic examination showed that by the fourth week of the experiment, radiolucent areas suggesting periapical lesions appeared in most of the specimens in the 6 groups (Figure 5B). Quantitative analysis of the apical shadow area imaging results (Figure 5D, a) showed that compared with the control group, the Fe_3_O_4_-CaO_2_-Hydrogel substantially suppressed the increase in the shadow area. There was no statistical difference between the Fe_3_O_4_-hydrogel and hydrogel groups, indicating that both groups had no inhibitory effect. The commonly used root canal disinfectant Ca(OH)_2_ paste showed a slight inhibitory effect; however, the effect was much lower than that of the Fe_3_O_4_-CaO_2_-Hydrogel. The specific values are provided in Table 2.

#### 2.8.2. Histological Analysis

Massive inflammatory cell infiltration can be observed in the lesion area (Figure 5C). In terms of the infiltration area, the Fe_3_O_4_-CaO_2_-Hydrogel group, CaO_2_-Hydrogel group and Ca(OH)_2_ paste group had significantly lower area compared with the control group (Figure 5D, b), the three groups showed a tendency to reduce inflammatory infiltration in the periapical tissue. Particularly for the Fe_3_O_4_-CaO_2_-Hydrogel group. The area of the periapical apical inflammatory infiltration zone of the Fe_3_O_4_-Hydrogel and control groups was relatively large. The specific values are provided in Table 3.

#### 2.8.3. In Vivo Safety

H&E staining showed that the Fe_3_O_4_-CaO_2_-Hydrogel did not damage the vital in-ternal organs of rats (Figure 6).

## 3. Discussion

Complete removal of root canal biofilm infection is still challenging [26]. To improve the success rate of root canal treatment, mechanical and chemical methods are often used to minimize bacterial infection. However, due to the root canal system’s complexity and the bacterial tolerance, achieving the desired effect is challenging; in particular, the nature of bacteria after biofilm formation makes it more difficult to remove than that in the planktonic state [27]. *E. faecalis* is one of the most frequently isolated bacteria in root canal treatment failure. Previous studies have revealed that 67–77% of *E. faecalis* are found in failed root canal treatment cases. At the same time, *E. faecalis* can endure long-term starvation and has strong adaptability to alkaline environment [28]. *S. sanguis* is often considered an opportunistic pathogen that plays an essential role in the early biofilm formation and bacterial adhesion stage. *S. sanguis* is also a component of root canal biofilms and one of the most commonly concerned microorganisms in the dental field. Recent studies have confirmed that both *E. faecalis* and *S. sanguis* are slightly resistant to disinfectants and antibiotics [29]. Therefore, the current experiment considered mainly *E. faecalis* and *S. sanguis* to obtain an infected root canal model.

The success of root canal treatment depends on many factors, of which the chemomechanical preparation of the root canal system and the control of microbial growth are the most critical factors. However, the anatomical complexity of the root canal system limits the efficiency of the clinical instrumentation, which complicates bacterial biofilm eradication, so the effectiveness of disinfectants becomes more critical for the elimination and prevention of root canal infection [30]. ROS have been widely used in recent years owing to their antibacterial effect. Photodynamic and sonodynamic therapy can induce ROS production; both therapies require oxygen consumption to generate ROS, further exacerbating the hypoxic environment and compromising the total effect of the antimicrobial therapy [31]. With advances in nanomaterials, iron-based nanozyme-mediated chemodynamic therapy (CDT) has gradually garnered increasing attention [1]; however, most CDTs applied for antibacterial effects require the use of exogenous H_2_O_2_, which causes irreversible damage to the normal tissues [32]. Self-supplying H_2_O_2_ modified nanozyme-loaded hydrogel can solve this problem, using CaO_2_ as the H_2_O_2_ donor [17,33,34], to generate ROS for antibacterial without using additional H_2_O_2_. To the best of our knowledge, this self-supplied H_2_O_2_ modified nanozyme has rarely been used for root canal disinfection. The carboxyl groups on HA can coordinate with Fe^3+^ or Ca^2+^, and they have a high affinity for Fe_3_O_4_ NPs and CaO_2_ NPs. Using HA, we integrated Fe_3_O_4_ NPs and CaO_2_ NPs into the modified nanozyme. HA also limits the increase in the nanoparticle size. SEM images showed that the average particle size of the synthesized modified nanozymes was approximately 30–40 nm; a smaller particle size results in higher reaction efficiency. Regarding the mixing ratio of the two nanoparticles, some researchers have used a much higher ratio of CaO_2_ NPs to Fe_3_O_4_ NPs to increase its antitumor effect [18,35], thus resulting in a high local H_2_O_2_ concentration; however, excessively high H_2_O_2_ concentration may decrease the nanozyme effectiveness [36]. Referring to the study of Li et al. [37], we used the same concentration ratio of CaO_2_ and Fe_3_O_4_ NPs.

The current study aimed to apply Fe_3_O_4_-CaO_2_-Hydrogel as a new disinfectant against the root canal bacterial biofilm. MTT assay, crystal violet staining, and live/dead staining demonstrated that the Fe_3_O_4_-CaO_2_-Hydrogel had a better inhibitory effect on the root canal biofilm metabolism. Noticeably, in previous studies, CaO_2_ was used as a root canal disinfectant. It exhibited a certain antibacterial effect at a high concentration [38] but was less effective than that of Fe_3_O_4_-CaO_2_-Hydrogel in this study at the same concentration. This is attributed to the difference in the antibacterial principle; the antibacterial principle of CaO_2_ is the H_2_O_2_ produced by the reaction with H^+^ [39]; however, the instability of the produced H_2_O_2_ makes it decompose faster in the bacterial environment, which interferes with the effect of CaO_2_ antibacterial effect. In the modified nanozyme-loaded hydrogel, the H_2_O_2_ generated by CaO_2_ are rapidly decomposed by Fe_3_O_4_ nanozymes into more toxic ROS, directly killing pathogens by damaging bacterial cell membranes and DNA. Meanwhile, the modified nanozyme-loaded hydrogel exhibited the same antibiofilm effect compared with the root canal disinfection strategy in which additional H_2_O_2_ was added in other studies [40]. The results of in vivo experiments showed that Fe_3_O_4_-CaO_2_-Hydrogel compared with the commonly used clinical root canal disinfection drug (Ca(OH)_2_ paste), significantly reduced the periapical lesion radiolucent areas (*p* < 0.05) (Figure 5D), suggesting that Fe_3_O_4_-CaO_2_-hydrogel can control root canal bacterial biofilm better than Ca(OH)_2_ paste. Ca(OH)_2_ paste releases hydroxyl ion, providing an alkaline pH, while *E. faecalis* is highly resistant to alkaline environments, so Ca(OH)_2_ paste is not an ideal disinfection material for root canal infection [41]. Root canal disinfection materials should exhibit both a good disinfection ability and biocompatibility [42]. Previous studies have shown that the high inflammatory activity of ROS might induce oxidative stress in native cells, posing a potential risk to cell viability [43]. However, the slow release of the ROS after the addition of the hydrogel can overcome this problem [44], avoiding the explosive increase in local ROS concentration. Hydrogel is a remarkable cross-linked 3D polymer network material. Recently, several thermosensitive hydrogels were developed. Their gelation and swelling changes can be achieved by increasing ambient temperature to physiological temperature, thus having a wide range of applications in the biomedical field. Currently, commonly used thermosensitive hydrogels can be divided into three systems, which are Poly (N-isopropylacrylamide) (pNIPAAm)-based systems, Poly(ethylene oxide)-b-poly(propylene oxide)-b-poly(ethylene oxide) (PEO–PPO–PEO)-based systems, and natural polymer systems [45]. For the pNIPAAm based system, many studies have focused on drug delivery in extreme environments, and its synthesis method is complicated; PEO-PPO-PEO-based systems have relatively fast dissolution rates and low mechanical strength [46]. The chitosan-based hydrogel in this study belongs to the natural polymer system hydrogel; it is easy to obtain and has good mechanical properties. The gelatin component in the selected natural polymer hydrogel acts as collagen to improve cell compatibility. CCK-8 assay results showed that even at a high concentration of the modified nanozyme, namely 1.5 mg/mL, cell viability was much higher than that reported in previous studies [18]. At the same time, the hydrogel played a role of sustained release; the ROS release curve (Figure 1L,M) showed that there was still ROS released from the hydrogel after 150 h, which is very important for long-term inhibition of bacterial biofilms. SEM images of the freeze-dried hydrogel also showed the characteristic of large-area pores of a representative CS-based hydrogel, which can serve as a scaffold for cell growth and provide a suitable environment for later tissue repair [47].

Furthermore, the root canal system comprising isthmus and lateral canals has a complex anatomical morphology [48]. Fe_3_O_4_-CaO_2_-Hydrogel at room temperature has good fluidity, which is very suitable for filling the isthmus of the root canal, type “C” root canal, apical ramification, and lateral root canal. After implantation into the root canal, the gelation was completed in 60 ± 5 s at 37 °C in the human body. The appropriate gelation time provided the convenience of clinical operation, allowing the hydrogel to fill the root canal and make close contact with the bacterial biofilm.

There are some limitations to this study, mainly in terms of inflammation control. The hydrogel has a better bactericidal effect; however, combined with the radiographic examination and histological analysis, it was observed that the hydrogel could only prevent the bacteria from continuing to destroy the periapical tissue, and it did not fully restore the pre-operative state of the tissue. Analysis of periapical tissue sections found that a certain number of inflammatory cells were aggregated in the root tip of the Fe_3_O_4_-CaO_2_-Hydrogel group. Although the number of inflammatory cells in the experimental group was generally less than that in the control group, the inflammation could not be completely inhibited; which may be attributed to the ROS released by the hydrogel. ROS play a key role in maintaining the M1 (pro-inflammatory) phenotype of macrophages due to the activation of the NF-κB signaling pathway involved in the expression of pro-inflammatory cytokines [49]; moreover, ROS increase the expression of IL-1β. ROS increased IL-1β expression, recruited neutrophils to the lesion, and formed clusters of neutrophils around it [50], which can adequately explain the histological findings. We believe that with the continuous consumption of ROS, the number of neutrophils decreases substantially and the periapical area tissue is further recovered. Theoretically, CaO_2_ in the hydrogel reacts to generate Ca(OH)_2_, which not only endows the hydrogel with continuous antibacterial ability but also causes a local increase in the concentration of Ca^2+^; the deviation between the concentration of extracellular Ca^2+^ and the physiological value activates intracellular signaling related to osteogenesis, enabling positive regulation of the osteogenic function of bone-forming cells [51,52]. Thus, our hydrogels exhibit potential osteogenic repair tissue function; however, further experiments are required to confirm this effect.

Although the modified nanoenzyme demonstrated good antibacterial properties and biocompatibility in this study, further studies should be conducted to evaluate the effect of this hydrogel on dentin hardness, color, and fracture resistance for translating preclinical studies of the material into advances in dental medicine.

Most research on antibacterial drugs is limited to the inhibition of bacteria. In the current mechanism of antibacterial treatment, the accumulation of ROS and metal ions can prevent bacterial growth but also triggers a pro-inflammatory response. In the future, we should focus on regulating the pro-inflammatory response of antibacterial materials to achieve a balance between the antibacterial and the anti-inflammatory effects. Ideally, antimicrobial materials should be able to quickly eliminate the bacterial biofilm while avoiding inflammation of the surrounding tissue.

## 4. Materials and Methods

The experiment was approved by the Institutional Animal Care and Use Committee of Jilin University (No. 20210607).

### 4.1. Synthesis and Characterization of the Modified Nanozyme

Synthesis steps for Fe3O4 NPs: First, ferric chloride (FeCl_3_ 6H_2_O) and ferrous chloride (FeCl_2_ 4H_2_O) were dissolved in deionized water at 60 °C at a molar ratio of 1:2; after stirring for 5 min, 25% ammonia solution was added dropwise. After 20 min, magnetic separation was performed to separate the product, which was then washed with deionized water several times and dried at 60 °C overnight. 

Synthesis steps for Fe_3_O_4_-CaO_2_ NPs: First, 0.4 g anhydrous calcium chloride and 150 mg hyaluronic acid (HA, Macklin, Shanghai, China) were added to 20 mL absolute ethanol and dissolved through ultrasonification; then, 2 mL of 25% ammonia solution and 3 mL of 30% hydrogen peroxide solution were added, stirred for 5 min., then centrifuged the mixture at 10,000 rpm for 10 min to collect the precursor product. This precursor product (2 mg/mL), along with Fe_3_O_4_ NPs (2 mg/mL), was dissolved in 20 mL absolute ethanol, stirred overnight, and centrifuged at 12,000 rpm for 10 min to obtain Fe_3_O_4_-CaO_2_ NPs. 

For transmission electron microscopy (TEM) characterization, the washed Fe_3_O_4_-CaO_2_ NPs were redispersed in ethanol; then the NPs were drop casted on carbon-coated copper grids. After evaporation of ethanol and water under ambient conditions, the obtained Fe_3_O_4_-CaO_2_ NPs was observed using a TEM (JEOL, Japan). The crystalline phase of CaO_2_ NPs was characterized using X-ray diffraction (XRD, X’Pert PRO MPD, the Netherlands). The chemical composition and valence states of elements were measured by X-ray photoelectron spectroscopy (XPS, Thermo Kalpha, Waltham, MA, USA).

### 4.2. Synthesis and Characterization of the Modified Nanozyme-Loaded Hydrogels

CS (1 g; Sigma, Burlington, MA, USA) was sterilized through UV lamp irradiation for 1 h on an ultra-clean bench and then dissolved in 50 mL of 0.1 mol/mL sterile hydrochloric acid aqueous solution (analytical grade, Beijing Chemical Factory, Beijing, China). The mixture was magnetically stirred overnight to form a CS solution. A GP solution was prepared by stirring 56% (*w*/*v*) β-glycerophosphate disodium hydrate (Sigma, Burlington, MA, USA) at room temperature (25 °C), and after its complete dissolution, it was filtered through a 0.22-μm filter for further use. A GA solution was prepared by stirring 0.5% (*w*/*v*) gelatin (Sigma, Burlington, MA, USA) at 55 °C; when no precipitation was observed, the suction was filtered through a 0.22-μm filter for further use.

5 mL of CS was added to a beaker, and then, 1 mL of GA and 0.15 mL of GP were added dropwise to the CS solution under an electronic stirrer continual stirring. After the solution was uniformly mixed, Fe_3_O_4_-CaO_2_ NPs were dissolved in 1 mL of 0.05 M NaOH solution by sonication and added dropwise to the above solution.

The gelation time was determined using the vial-tilting method [53]; after adding Fe_3_O_4_-CaO_2_-Hydrogel in a vial, the sol-to-gel transition of the mixture at 37 °C was determined by vial tilting every 10 s. The gelation time was recorded when the hydrogel formed and stopped flowing upon vial tilting. SEM and SEM mapping measurements were conducted by a scanning electron microscope (SEM, JSM-6700F, JEOL, Tokyo, Japan).

### 4.3. Detection of ROS Release from the Hydrogels

The release of ROS from the hydrogels at different time intervals was analyzed using a diphenylisobenzofuran (DPBF) probe. To evaluate the effect of concentration, different concentrations of modified nanozymes (0.5 mg/mL, 1.0 mg/mL, 1.5 mg/mL, 2.0 mg/mL, and 2.5 mg/mL) were loaded in the hydrogel. To simulate normal and bacterial environments, the hydrogel samples were treated with phosphate buffered saline (PBS) at pH 7.4 and pH 5.5, respectively. After different periods of leachate solution reacted with the DPBF probe, it was transferred to a 96-well plate, the generation of ROS was demonstrated by the characteristic absorption decrease in the DPBF at 420 nm by a UV–vis absorption spectrum.

### 4.4. Concentration Selection of the Modified Nanozyme

Biocompatibility and the antibacterial effect were examined using a cell counting kit-8 (CCK-8) assay and a colony-forming unit (CFU) counting method, respectively, to determine the optimal concentration of the modified nanozyme in the hydrogel.

#### 4.4.1. CCK-8 Assay

The modified nanozymes with different concentrations (0.5, 1.0, 1.5, 2.0, and 2.5 mg/mL) were loaded into the hydrogel, and cell toxicity was assessed using the extracts of hydrogel samples. Cell culture without the extract was used as a control. Briefly, mouse fibroblasts (L929) were cultured in a 96-well plate for 24 h, with 2000 cells per well, and 5 replicate wells were set in each group. After 24 h, the medium was replaced with a leaching solution. After 3 days, the CCK-8 kit reagent was added, and cytotoxicity was analyzed at 450 nm using a microplate reader (SYNERGY HT, BIO-TEK, Winooski, VT, USA).

#### 4.4.2. CFU Count Test

The loading concentration of modified enzymes was the same as that in the Section 4.4.1; the bacterial culture without hydrogel was used as the control group. *Enterococcus faecalis* (ATCC-29212) and *Streptococcus sanguis* (ATCC-10556) were cultured on cover glass for 7 days to form biofilms, the initial concentration of the two strains was 1 × 10^7^ CFU/mL, and 1 mL of hydrogel was co-cultured with the cover glass for 24 h. Then, the biofilms were collected from the cover glass by mechanical scraping and ultrasonic or turbine vibration. Collected *E. faecalis* biofilms was inoculated on brain–heart infusion (BHI, Oxoid, Basingstoke, UK) broth agar and collected *S. sanguis* biofilms was inoculated on Columbia blood plate. Then, all plates were incubated anaerobically at 37 °C for 24 h, and the number of colonies formed on each plate was counted.

### 4.5. In Vitro Antibiofilm Test

Five groups were used in the experiments, namely, Fe_3_O_4_-CaO_2_-Hydrogel; CaO_2_-Hydrogel; Fe_3_O_4_-Hydrogel; Hydrogel and Control group. Bacterial cultures were not treated with the hydrogel in the control group.

#### 4.5.1. Effect of the Modified Nanozyme Hydrogel on Single-Strain Biofilms

CFU counting experiments were used to evaluate the antibiofilm ability of the hydrogels against individual bacterial species. Briefly, the biofilm samples were prepared in the same way as in Section 4.4. After 24 h of anaerobic contact with the samples, each group of hydrogels was scraped and sonicated or vortexed to collect bacteria. Bacterial liquid (100 μL) was taken from the well, inoculated in a solid medium, and CFU counts were obtained after 24 h of anaerobic culture on all plates.

#### 4.5.2. Effects of the Modified Nanozyme Hydrogel on Multi-Strain Biofilms

The initial concentrations of *E. faecalis* and *S. sanguinis* were adjusted to 1 × 10^7^ CFU/mL, and the volume ratio was 1:1 in 24-well plates. The cover glass was pre-placed at the bottom of each well, and the bacterial culture medium was BHI with 2% sucrose. After seven days of culture, a light-yellow film was formed on the surface of the cover glass, indicating that the biofilm had been successfully cultivated. 

For bacterial live/dead staining experiments, after 24 h of co-incubation of cover glass with different samples. The SYTO 9 dye and propidium iodide (PI) dye were mixed in a 1:1 ratio and diluted with normal saline to make the stain. Each cover glass was stained for 30 min and observed by confocal laser scanning microscopy (CLSM), in imaging, green indicated the location of live bacteria, red indicated dead bacteria, and orange indicated areas where there was a large overlap between live and dead bacteria.

3-(4,5-dimethyl-thiazol-2-yl)-2,5-diphenyltetrazolium bromide (MTT) assay was used to evaluate the metabolic activity of the biofilms. The as obtained cover glass was transferred to a new well plate, and the hydrogels and biofilms of different experimental groups were co-cultured for 24 h (the medium was BHI). Then, the cover glass was taken out and treated with MTT dye (1 mL) and incubated in darkness at 37 °C for 1 h in 5% CO2; the formazan crystals were dissolved with an equal amount of DMSO and shaken horizontally in a shaker at 100 rpm for 30 min. Then, the formazan crystals were transferred to a new 96-well plate, and a microplate reader was used to read the absorbance at 540 nm.

For the crystal violet staining assay, after co-culturing with the hydrogel for 24 h, the cover glass was taken out, immersed in a glutaraldehyde solution (1 mL) for 30 min, and then treated with the same amount of crystal violet solution instead of glutaraldehyde. The remaining biofilms were combined with crystal violet for 20 min, decolorized with 95% alcohol, transferred to a 96-well plate, and then, the OD values were recorded at 600 nm.

### 4.6. Effect of the Modified Nanozyme Hydrogel on Bacterial Biofilm Structure

SEM was used to observe the effect of hydrogel on bacterial cell membranes, and these findings were compared with normal bacterial morphology to observe the degree of cell membrane rupture in *E. faecalis* and *S. sanguis*. The two types of bacteria grown on the cover glass were co-cultured with each group of hydrogels for 24 h, fixed with 4% paraformaldehyde for 10 h at 4 °C, dehydrated with alcohol gradient, dried, and then sprayed with gold for observation. K ions are crucial for bacterial metabolism. When the permeability of the membrane changes, the concentration of K^+^ ions outside the bacteria increases. The change in K ion concentration is an essential indicator for determining the integrity of the membrane. With sterile BHI as the control group, the bacterial liquid after hydrogel treatment in each group was centrifuged at 5000 rpm for 3 min, collect supernatant, and K^+^ ion content in the samples was detected through inductively coupled plasma (ICP) emission spectrometry.

### 4.7. Exopolysaccharide Detection

In accordance with the Albuquerque’s method [54] for extracting exopolysaccharides from cell membranes, the biofilms were cultured in the same manner as described in Section 4.4. The cover glass was separately immersed in the hydrogel groups for 24 h and then taken out. The residual colloid on the surface was gently washed with PBS, and the residual biofilm was collected by mechanical scraping, ultrasonication, or vortexing, and centrifuged at 4 °C and 12,000 rpm for 3 min. The supernatant was transferred to a 96-well plate, and then, to each well, 40 μL distilled water was added, followed by the addition of 40 μL of 6% phenol and 200 μL of 97% concentrated sulfuric acid; the OD values were recorded at 490 nm after stand for 20 min.

### 4.8. In Vivo Experiments

To observe the ability of the modified nanozyme-loaded hydrogel to remove the biofilm in vivo, 8-week-old Sprague–Dawley (SD) male rats (200–250 g) were used to establish a root canal infection model. Six groups were used, namely, Fe_3_O_4_-CaO_2_-Hydrogel; CaO_2_-Hydrogel; Fe_3_O_4_-Hydrogel; Hydrogel; Ca(OH)_2_ paste group; and Control group without hydrogel treatment.

#### 4.8.1. Preparation of the Root Canal System

A rat mandibular first molar distal root canal was used to establish the infection model, and sodium pentobarbital was injected into the abdominal cavity for general anesthesia. A sterile rose head carbide bur attached to a contra-angle high-speed handpiece was used with sufficient water coolant to penetrate and remove the roof of the pulp chamber; the pulp was removed using a barbed broach. Root canals were enlarged till size 10 with k hand files. After rinsing and sterilizing with 5.25% NaClO and 17% ethylene diamine tetraacetic acid (EDTA), temporarily closed with a glass ionomer.

#### 4.8.2. Intracanal Infection

After one week, the glass ionomer was removed. The concentrations of *E. faecalis* and *S. sanguis* were adjusted to 1 × 10^9^ CFU/mL, and the volume ratio was 1:1, a total of 10 µL; the bacterial solution was added to the root canal with ^#^6 K-stainless steel hand file.

#### 4.8.3. Intracanal Medication

One week after the bacteria were planted, the temporary sealing material was removed, and a ^#^6 K-stainless steel hand file was used to assist each group of hydrogels to reach the root canal; this was again temporarily sealed with the glass ionomer.

#### 4.8.4. Radiographic and Histological Analyses

X-ray films were taken at 0, 2, and 4 weeks to observe the size of the apical trans-mission shadow area. Tissue staining was performed to analyze the aggregation of inflammation. Thereafter, the rats were sacrificed. The mandibles were taken out, fixed in 4% paraformaldehyde for 1 week, immersed in 10% EDTA for decalcification for 2–3 months, embedded in paraffin, buccolingually sliced, and subjected to hematoxylin-eosin (H&E) staining.

#### 4.8.5. In Vivo Safety

After the experiment, the SD rats were sacrificed. The main organs of the rats were then harvested, fixed in buffered formalin, embedded in paraffin, sectioned, and stained with H&E.

### 4.9. Statistical Analysis

The statistical analysis was obtained by the software GraphPad Prism8 (GraphPad, San Diego, CA, USA). All data are expressed as Mean ± Standard Deviation (MD). The unpaired two-tailed Student’s t-test was used for statistical comparisons between the two groups. Multiple group comparisons were performed using a one-way analysis of variance. * *p* < 0.05, ** *p* < 0.01, *** *p* < 0.001 and **** *p* < 0.0001. ns, not significant. All experiments were performed in triplicate and repeated at least three times.

## 5. Conclusions

We synthesized a self- supplying H_2_O_2_ modified nanozyme and embedded it into the hydrogel at a suitable concentration. The modified nanoenzyme-loaded hydrogel effectively inhibited the growth of root canal biofilms in vivo as well as in vitro. It also decreased the metabolic activity without affecting biocompatibility, and the antibacterial properties of ROS did not make the bacteria resistant, suggesting that the hydrogel synthesized in the present study could be developed into a new root canal disinfectant or sealant in the future.

## Figures and Tables

**Figure 1 ijms-23-10107-f001:**
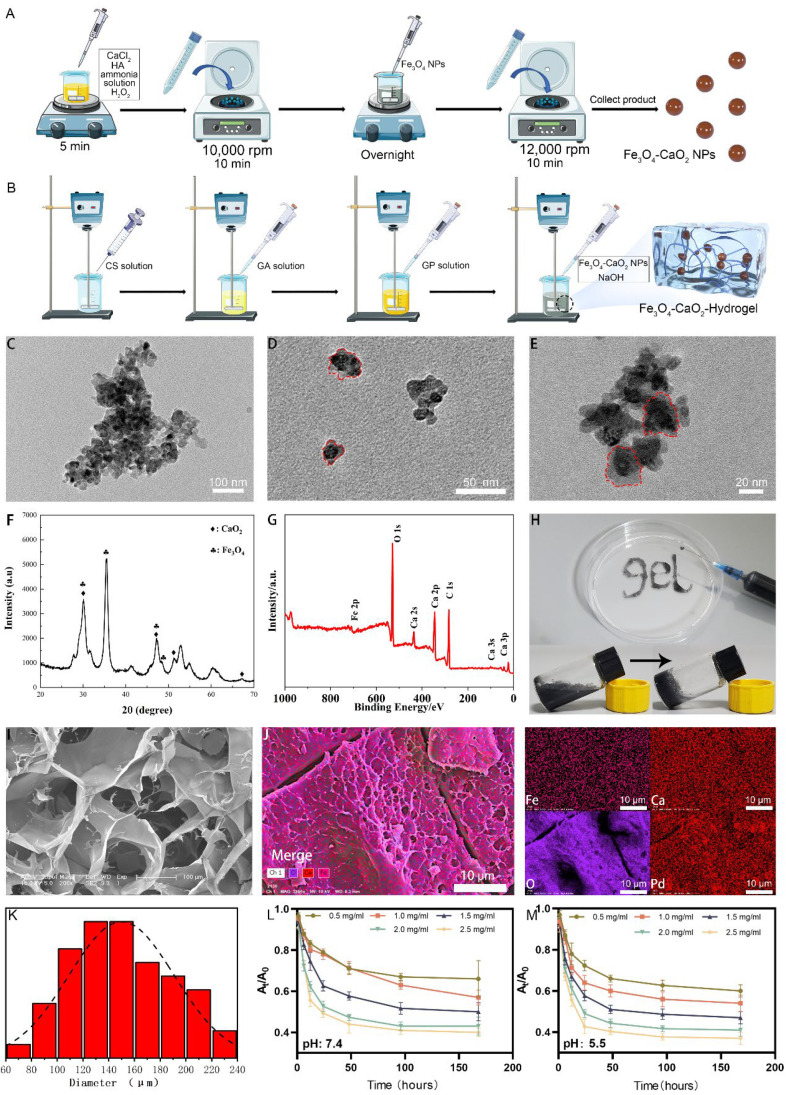
Characterization of Fe_3_O_4_-CaO_2_ NPs and Fe_3_O_4_-CaO_2_-Hydrogels. (**A**) Schematic illustration of the preparation of Fe_3_O_4_-CaO_2_ NPs. (**B**) Schematic illustration of the preparation of Fe_3_O_4_-CaO_2_-Hydrogel. (**C**–**E**) Representative TEM images of the prepared Fe_3_O_4_-CaO_2_ NPs, Red circles represent Individual Fe_3_O_4_-CaO_2_ NPs. (**F**) XRD, (**G**) XPS analysis of the prepared Fe_3_O_4_-CaO_2_ NPs. (**H**) Demonstration of injectable property of Fe_3_O_4_-CaO_2_-Hydrogel, Inset: Gelling process of Fe_3_O_4_-CaO_2_-Hydrogel. (**I**) SEM image of the Fe_3_O_4_-CaO_2_-Hydrogel. (**J**) Elemental mappings of Fe, Ca, O, and Pd of Fe_3_O_4_-CaO_2_-Hydrogel. (**K**) Pore size distribution of the Fe_3_O_4_-CaO_2_-Hydrogel. (**L**,**M**) The release of ROS by different dosages of Fe_3_O_4_-CaO_2_ NPs embedded in hydrogels under different pH values.

**Figure 2 ijms-23-10107-f002:**
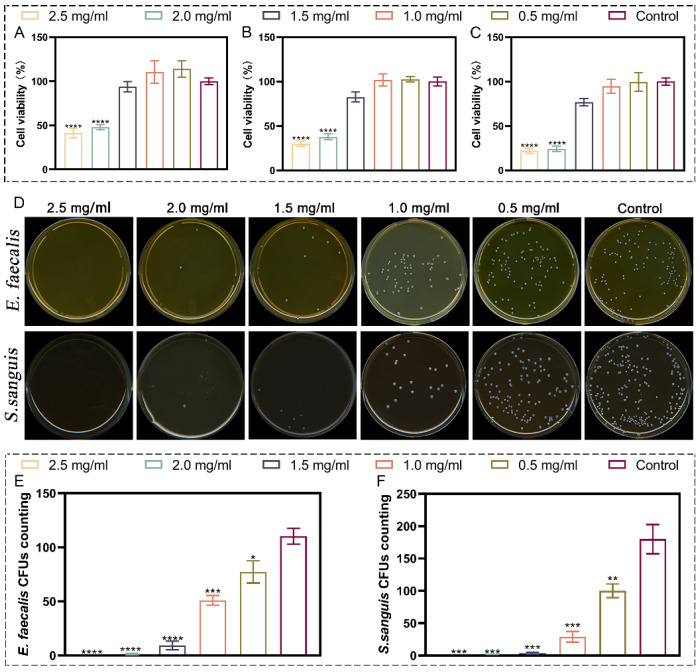
In vitro cytocompatibility test and CFU count test. (**A**–**C**) CCK-8 assay of L929 cells on days 1–3. (**D**) Typical photographs of *E. faecalis* and *S. sanguis* bacterial colonies after cocultivation with hydrogel samples containing different concentrations of Fe_3_O_4_-CaO_2_ NPs. (**E**,**F**) Statistical results related to the bacterial colony numbers of *E. faecalis* and *S. sanguis*. * *p* < 0.05, ** *p* < 0.01, *** *p* < 0.001, and **** *p* < 0.0001.

**Figure 3 ijms-23-10107-f003:**
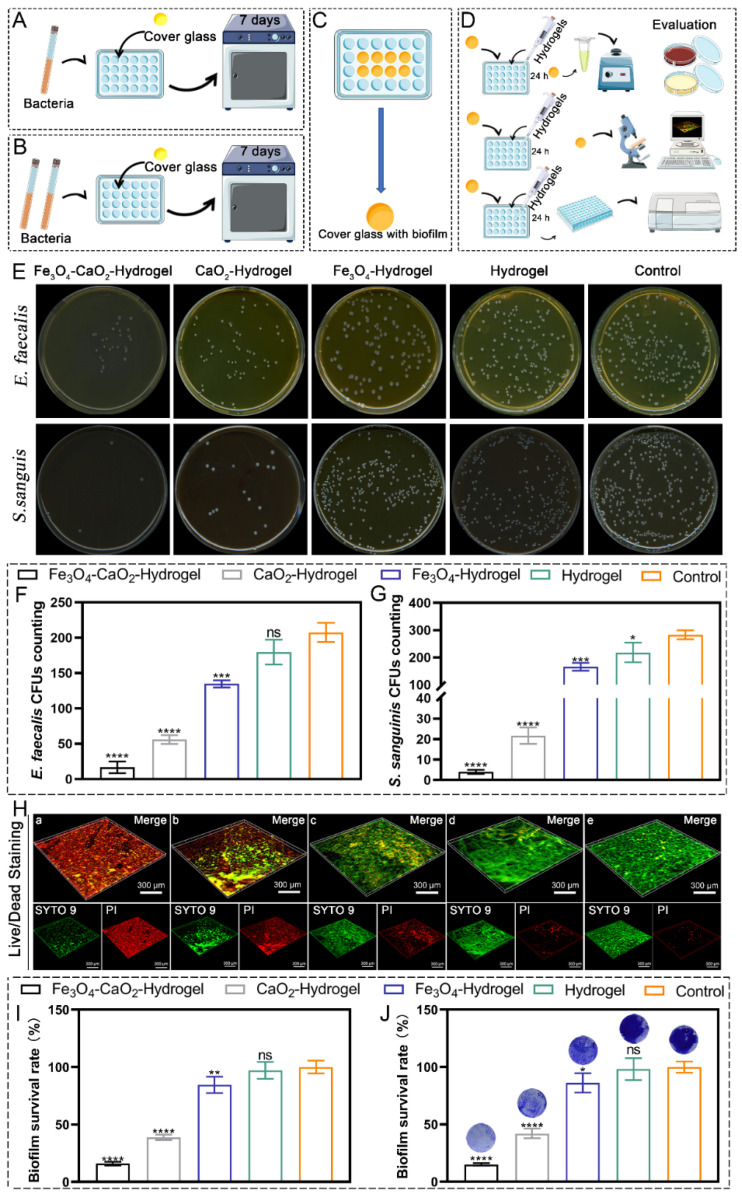
In vitro antibacterial tests. (**A**) Formation of single-strain biofilms. (**B**) Formation of multi-species biofilms. (**C**) Biofilm formation on the surface of the cover glass. (**D**) Antibiofilm Evaluation of Hydrogels. (**E**) bacterial colony plate counting. (**F**,**G**) The corresponding quantitative analysis according to colony plate counting. (**H**) Live/Dead staining of bacterial biofilm, live bacteria were stained by SYTO 9, resulting in green fluorescence. Dead bacteria with damaged cell membranes were stained by PI, resulting in red fluorescence. a: Fe_3_O_4_-CaO_2_-Hydrogel, b: CaO_2_-Hydrogel. c: Fe_3_O_4_-Hydrogel. d: Hydrogel. e: Control. (**I**) MTT assay measuring the bacterial biofilm activity in different groups. (**J**) Crystal violet staining assay measuring the bacterial biofilm activity in different groups, Inset: Staining status of the cover glass after treatment in different groups. * *p* < 0.05, ** *p* < 0.01, *** *p* < 0.001, **** *p* < 0.0001. ns, not significant.

**Figure 4 ijms-23-10107-f004:**
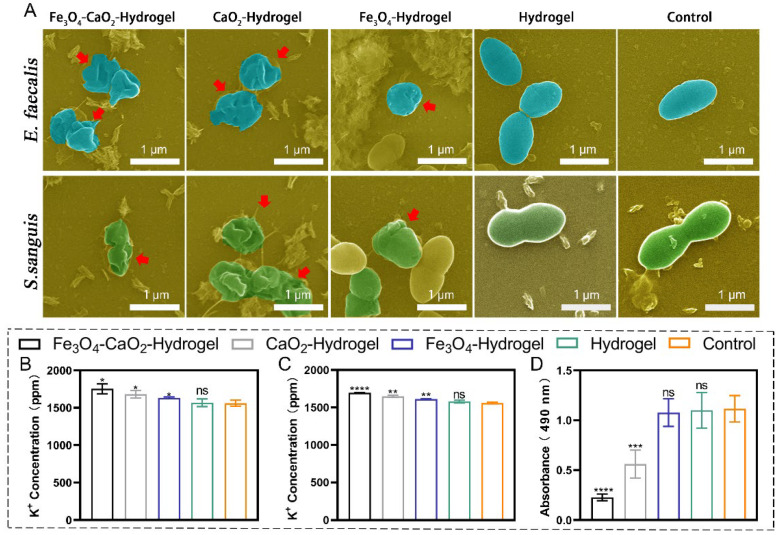
Antibacterial mechanism of Fe_3_O_4_-CaO_2_-Hydrogel. (**A**) SEM observation, red arrows show apparent damage and deformation of bacteria. ICP assay of (**B**) *E. faecalis* and (**C**) *S. sanguis*. (**D**) EPS production after co-culture of bacterial biofilms with each group of samples. * *p* < 0.05, ** *p* < 0.01, *** *p* < 0.001, **** *p* < 0.0001. ns, not significant.

**Figure 5 ijms-23-10107-f005:**
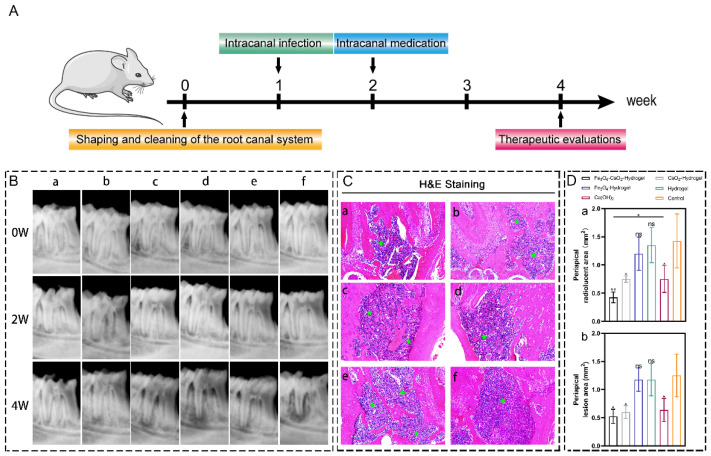
(**A**) Schematic diagram of in vivo experiment results. (**B**) After root canal preparation (0 week), intracanal medication (2 weeks), and postoperative (4 weeks) radiographs. a: Fe_3_O_4_-CaO_2_-Hydrogel, b: CaO_2_-Hydrogel. c: Fe_3_O_4_-Hydrogel. d: Hydrogel. e: Ca(OH)_2_ paste f: Control. (**C**) Representative images of H&E staining for the a: Fe_3_O_4_-CaO_2_-Hydrogel, b: CaO_2_-Hydrogel. c: Fe_3_O_4_-Hydrogel. d: Hydrogel. e: Ca(OH)_2_ paste f: Control. (Asterisk: dense inflammatory infiltration zone). (**D**) Evaluation of apical lesions: a: comparison of mean apical radiolucent area in X-ray images. b: comparison of the apical lesion area in H&E-stained tissue sections. * *p* < 0.05, ** *p* < 0.01. ns, not significant.

**Figure 6 ijms-23-10107-f006:**
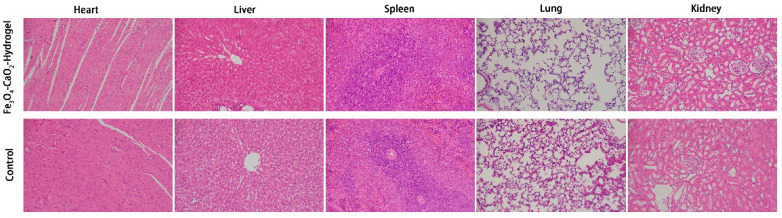
H&E stained images of major organs (heart, liver, spleen, lung, and kidney) collected from the Fe_3_O_4_-CaO_2_-Hydrogel group and the Control group.

**Table 1 ijms-23-10107-t001:** Specific values for crystal violet staining.

Groups	Mean ± SD	*p* Value
Fe_3_O_4_-CaO_2_-Hydrogel	0.12 ± 0.01	*p* < 0.0001
CaO_2_-Hydrogel	0.34 ± 0.04	*p* < 0.0001
Fe_3_O_4_-Hydrogel	0.70 ± 0.06	*p* < 0.05
Hydrogel	0.80 ± 0.08	*p* > 0.05

**Table 2 ijms-23-10107-t002:** Specific data of each group of radiolucent areas.

Groups	Mean ± SD	*p* Value
Fe_3_O_4_-CaO_2_-Hydrogel	0.43 ± 0.10	*p* < 0.01
CaO_2_-Hydrogel	0.75 ± 0.06	*p* < 0.05
Fe_3_O_4_-Hydrogel	1.20 ± 0.29	*p* > 0.05
Hydrogel	1.35 ± 0.31	*p* > 0.05
Ca(OH)_2_	0.75 ± 0.24	*p* < 0.05

**Table 3 ijms-23-10107-t003:** Specific data of each group of periapical apical inflammatory infiltration zone.

Groups	Mean ± SD	*p* Value
Fe_3_O_4_-CaO_2_-Hydrogel	0.53 ± 0.13	*p* < 0.05
CaO_2_-Hydrogel	0.60 ± 0.12	*p* < 0.05
Fe_3_O_4_-Hydrogel	1.18 ± 0.21	*p* > 0.05
Hydrogel	1.18 ± 0.29	*p* > 0.05
Ca(OH)_2_	0.64 ± 0.21	*p* < 0.05

## Data Availability

Data available on request due to restrictions, e.g., privacy or ethical. The data presented in this study are available on request from the corresponding author.

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
