# Peer review of "A Self-Supplying H2O2 Modified Nanozyme-Loaded Hydrogel for Root Canal Biofilm Eradication"

_ijms, 2022, doi:10.3390/ijms231710107_

Round 1
Reviewer 1 Report
Dear Authors,
you made a really great work!
However, minor improvements are required before acceptance.

Reviewer 2 Report
The manuscript under review attempts to evaluate a self-supplying H2O2 modified nanozyme-loaded hydrogel used to eradicate root canal biofilm. In general, the manuscript did capture details of the study design and implementation of the project. All the sections of the manuscript are well written and concluded. The study should be of sound design, clear practical and clinical interest, and I suggest revision before considering. Kindly find below the comments which will help the authors to revise the manuscript.
Abstract:
1. Kindly provide structured abstract, or cover the background/aim, methods, results and conclusion as a paragraph.
2. Kindly mention the significant values
3. Kindly provide MesH keywords
Introduction:
1. Line 47: ROS- complete form in abstract and main should be mentioned separately; hence kindly note complete form once it appears first in the main text followed by short form.
2. Line 59: correction c arrier as the carrier.
3. Kindly mention the aim/objective of the study in the last
Material and methods:
1. Kindly shift materials and methods after the introduction and before the results, as the authors mentioned several findings and steps which need to be referred to in the methods. Hence advised to shift it before the results.
2. Kindly provide the ethical approval details at the start of materials and methods.
3. Why did the authors prefer the 30–40 nm diameter range?
4. vial-tilting method: kindly provide a reference.
5. kindly illustrate the methodology so that it will be reader easy
Results:
1. Results are well presented.
2. Kindly provide the tables and significant values (P values)
Discussion:
1. Kindly discuss the objectives of the studies and compare them with the literature.
2. Importance of root canal irrigants and Self-supplying H2O2 Modified Nanozyme-loaded Hydrogel.
3. Recent updates in hydrogels.
4. Clinical implementations.
5. Write the limitations of the study.
6. Future recommendations and directions, apart from the future studies to be conducted.
Round 2
Reviewer 2 Report
Dear Authors,
The authors have addressed all the comments and responded perfectly; the manuscript is much improved, and I suggest authors to added significant p-values in the abstract. I would like to congratulate the authors and appreciate their hard work. wishing you all the best in future endeavours.
Best regards
